# *"It would be better for those of us who have the disease not to be ashamed":* Insights from people living with chronic hepatitis B virus infection and healthcare workers providing HBV care in Kilifi, Kenya

Louise O. Downs[1,2]*, Juliet Odhiambo[2], Mwanakombo Zaharani[2], Oscar Chirro[2], Benson Safari[2], Janet Seeley[3,4], Philippa C. Matthews[5,6,7], Nadia Aliyan[8], Nancy Kagwanja[2]

1 Nuffield Department of Medicine, University of Oxford, Oxford, United Kingdom, 2 KEMRI-Wellcome Trust Research Programme, Kilifi, Kenya, 3 London School of Hygiene and Tropical Medicine, London, United Kingdom, 4 Africa Health Research Institute, KwaZulu-Natal, South Africa, 5 The Francis Crick Institute, 1 Midland Road, London, United Kingdom, 6 Division of Infection and Immunity, University College London, London, United Kingdom, 7 University College London Hospital, London, United Kingdom, 8 Kilifi County Referral Hospital, Kilifi, Kenya

* Louise.downs@exeter.ox.ac.uk

## Abstract

Chronic hepatitis B infection (CHB) causes over 1 million deaths annually, with a large burden of morbidity and mortality in the WHO-African Region (WHO-AFRO) where <5% of people are diagnosed and 0.2% are on treatment. Studies have shown that understanding of hepatitis B virus (HBV) here is often poor, and people living with HBV (PLWHB) can experience stigma and discrimination. However there has been little documentation on the impact of an HBV diagnosis on the lives of PLWHB in the WHO-AFRO region or community involvement in improving care provision. We undertook two focus group discussions (FGDs) with PLWHB and two with healthcare workers (HCWs) providing HBV care at Kilifi County Referral Hospital (KCRH), Kenya to explore experiences of living with HBV and barriers to accessing care. FGDs were conducted primarily in Kiswahili, transcribed verbatim and translated into English. The data were analysed thematically using NVivo version 14. PLWHB and HCWs at KCRH had a good understanding of HBV which was likely influenced by a concurrent research study on HBV, however they reported low awareness in the general community, and there is no local name for the infection. Many PLWHB were shocked at their initial diagnosis with mixed reactions from friends and family. Costs of transport and concerns about lost employment were the biggest barriers to care. Many people suggested decentralised clinics would reduce loss to follow up, however others would rather be treated far from home to preserve anonymity. Stigma was highlighted as a major issue, leading to feelings of isolation, rejection and discrimination. Community education, wider testing and advocacy by well-respected community members were

**Data availability statement:** Data provided in the paper, including illustrative quotes, can be used without request but with full reference to the article. We are unable to make the interview transcripts more freely available because the qualitative data reported in this article contains potentially identifiable information for individuals, health facilities, and organisations. The ERC-approved protocol and participant consent forms did not include obtaining explicit consent regarding the possible use of anonymised data in the public domain via a data repository. Questions about data access can be directed through the Data Governance Committee of the KEMRI/Wellcome Trust Research Programme at DataGovernancecommittee@kemri-wellcome.org.

**Funding:** This research was funded in whole or in part by the Wellcome Trust (grant number 225485/Z/22/Z). For the purpose of Open Access, the author has applied a CC-BY public copyright license to any author accepted manuscript version arising from this submission. LOD is funded by a Wellcome Trust Grant (number 225485/Z/22/Z) and Oxford University John Fell Fund (award number 0012112). PCM receives core funding from the Francis Crick Institute (Ref: CC2223) and from University College London Hospitals NIHR Biomedical Research Centre (BRC). JO receives core funding from KWTRP. The funders had no role in study design, data collection and analysis, decision to publish, or preparation of the manuscript.

**Competing interests:** PCM has previously received funding from GSK for a PhD student in her group. This has had no influence over the planning or undertaking of the study, the writing of the manuscript or decision to publish.

mentioned as key methods to reduce HBV transmission. Decentralisation of clinics may improve access to care; however, this needs to be developed in careful consultation with PLWHB to ensure they are acceptable and accessible to all.

## Introduction

Approximately 256 million people worldwide are living with chronic hepatitis B infection (HBV) with the highest burden of infection being in the WHO-African Region (WHO-AFRO) [1]. HBV has been neglected in terms of healthcare provision, education, research and policy [2,3] and it is estimated only 4.2% of those living with HBV in WHO-AFRO have been actually diagnosed. Knowledge around HBV both in communities and healthcare facilities is often limited [1,4]. In Kenya, an estimated 3–5% of the population is living with HBV [5], but screening is largely restricted to those living with HIV, blood donors, dialysis patients and in some locations antenatal screening. There is minimal population screening and vaccine is mostly unavailable to adults at risk of exposure (such as healthcare workers) who were not vaccinated as children. Elimination efforts in Kenya are gaining traction with the Ministry of Health working on new HBV management guidelines and a National Strategic Plan. As these efforts evolve, it is critical to incorporate both the voices of people living with HBV (PLWHB) and the perspectives of healthcare workers (HCWs), who face operational and systemic challenges in delivering care.

These themes are echoed across the broader WHO-AFRO region, where emerging reports highlight the psychosocial burden of an HBV diagnosis and the pervasive stigma experienced by PLWHB [6,7]. The World Hepatitis Alliance has piloted a regional HBV stigma survey to better understand these challenges [8]. Key findings include PLWHB reporting being avoided at work and in public spaces, feeling they are treated unfairly in healthcare settings and being worried about disclosing their status to families. Representation of those with lived experience will improve advocacy, and inform service provision, driving progress towards global elimination targets [9]. The positive role of peer support workers in HBV care is being increasingly recognised, to encourage testing, improve treatment adherence and reduce stigma [10]. Healthcare workers (HCWs) providing care for PLWHB have a unique insight into barriers to long-term follow-up and medication. They understand the healthcare infrastructure for PLWHB, have insights into gaps and challenges, and can advise how access could be improved.

We aimed to describe the experiences of PLWHB and HCWs caring for PLWHB in Kilifi County Referral Hospital (KCRH) in Kenya, to help inform care delivery. We also sought to establish a scalable platform for qualitative data collection that would be useful in other settings. We convened focus group discussions (FGDs), aiming to i) explore understanding of HBV infection on an individual and community level, ii) discuss how the lives of PLWHB have changed since their diagnosis and iii) investigate barriers to receiving care and consider how these could be reduced.

## Materials and methods

The methods and results of this study are reported in compliance with the Standards for Reporting Qualitative Research (SRQR) [11].

### i) Study context

We conducted FGDs with both PLWHB and HCWs providing HBV services. This study was conducted at KCRH, a referral facility in Coastal Kenya, servicing an area of around 12 000 km² and a population of 1.5 million people [12]. KCRH has 300 inpatient beds and sees 193 000 outpatients every year. Poverty levels in Kilifi are above the national average for Kenya, with 53% of the population living in poverty in 2022 compared to the national average of 40% [13]. At the time of this study, HBV care at KCRH was provided at the Comprehensive Care Clinic (CCC). This is a secondary care outpatient clinic only caring for people living with HIV and HBV. The CCC provides adherence counselling, medication and follow up for both adults and children living with HBV. At KCRH, further laboratory or imaging assessment to determine HBV treatment eligibility is either not available (such as HBV viral load and elastography measurement) or beyond the financial means of most PLWHB (such as complete blood count or liver function tests to calculate the aspartate to platelet ratio index (APRI score)), therefore standard care is that all those testing HBsAg positive receive nucleoside analogue (NA) therapy using tenofovir/lamivudine (TDF/3TC) combination therapy. This approach is in line with conditional recommendations for 'treat all' which are incorporated in new WHO guidelines released in 2024 [14].

### ii) Data collection

We conducted four FGDs, two with PLWHB (consisting of six men and seven women) on 18th June 2024, and two with HCWs (consisting of seven men and seven women) on 14th November 2024. PLWHB were invited to participate in FGDs during peer support educational sessions initiated by the CCC matron when coming to pick up medication, and HCWs were invited via a group messaging platform for CCC staff. Participants were purposively sampled to create groups with diverse ages and experience, and in the case of HCWs to ensure a spread of cadres involved in HBV service delivery.

FGDs were held in private rooms in the CCC to promote participant privacy. Participants were given a number from 1-6 or 1–7 within their group to enhance anonymity of the respondents. FGDs with PLWHB were conducted primarily in Kiswahili by MZ, OC, and BS and facilitated by JO, a member of the Community Liaison Group from KEMRI Wellcome Trust Research Programme (KWTRP) experienced in qualitative data collection. FGDs with HCWs were conducted in both English and Kiswahili by MZ and JO, facilitated by LD. To minimise the risk of interviewer influence on FGDs, we used semi-structured topic guides (S1 File and S2 File) to direct FGDs, informed by study objectives and literature about patient experiences of accessing and receiving HBV care. The topic guides covered the understanding of HBV, reactions to diagnosis, the impact on the lives of PLWHB, barriers to care and how these could be reduced. Staff leading discussions were trained to employ open ended questions ensuring participants could freely express their views without steering. We encouraged personal reflection on experiences to reduce the influence of previous participation in research and peer support groups, rather than generalised knowledge. We triangulated findings between both PLWHB and HCWs to identify recurring themes and reduce reliance on previously shared narratives. FGDs took around 1.5 hours each, were audio recorded using an encrypted dictation device from KWTRP, and those participating were given refreshments. PLWHB attending specifically for the FGDs were compensated lost earnings of around USD $5, as is standard for community engagement activities at KWTRP.

### iii) STRIKE-HBV partner study

This work was undertaken in collaboration with the STRIKE-HBV study, a research study funded by the Wellcome Trust, based at KWTRP. STRIKE-HBV offered free HBV testing to all those attending KCRH between March 2023 – May 2024,

and free liver health assessment to anyone living with HBV as previously described [15]. STRIKE-HBV enrolled 102 PLWHB, including those newly diagnosed and those already known to be living with HBV and engaged in care. Activities undertaken by STRIKE-HBV including the status of HBV care in KCRH are detailed in S1 Table and educational material developed for STRIKE-HBV is available online [16]. All PLWHB who took part in these FGDs were also enrolled in STRIKE-HBV.

### iv) Data Analysis

Following FGDs, audio files were transcribed verbatim and translated into English and analysed in Nvivo version 14. Both inductive and deductive thematic analysis [17] was used to scrutinise the data to identify and analyse patterns and draw out themes using constant comparison technique [18]. Transcripts from all FGDs were examined on a line-by-line basis and the coding framework was first developed by LD and iteratively updated as new data came in. Sections with unclear themes were examined collaboratively with the research team, and differences in interpretation were resolved through group consensus to ensure consistency and strengthen the rigour of the analysis. Codes were developed based on the original questions asked by the interviewers and areas of interest to the authors, along with new themes being identified 'a priori' based on existing literature and tailored to the local context [4,19]

### v) Ethics and governance

This study was approved by the Scientific Ethics Review Unit (SERU) (SERU 3416). Written consent was obtained from all FGD participants prior to participation in FGDs, including consent for audio recording.

## Results

Focus groups with PLWHB included six men and seven women with a median age of 37 years (range 24 – 61). Five participants had been newly diagnosed with HBV in the STRIKE-HBV study, eight already knew their diagnosis prior to enrolment in STRIKE-HBV. Two of those participants who were newly diagnosed had family members already known to be living with HBV. HCW focus groups consisted of nine women and five men, with a median age of 31 years (range 18 – 61). The range of cadres included three clinical officers, three community health promoters (CHPs), two nurses and then one of each: counsellor, adherence supervisor, peer educator, nutritionist, records officer, and laboratory technician. A summary of the findings is shown in Table 1.

### i) Knowledge and understanding of HBV

**Understanding HBV transmission and treatment.** Both PLWHB and HCWs knew HBV is a virus causing liver disease and is transmitted sexually or by contact with infected blood. Sexual transmission was mentioned most by both groups, including men and women, and still considered the most common route of transmission. As participating PLWHB were those already receiving treatment, they were aware of HBV medication, and some HCWs specifically mentioned the use of tenofovir/lamivudine combination therapy. In both groups however, there was still some confusion around what happens after taking treatment for 6 months - some PLWHB thought they would be cured; however, others knew this was not the case:

*"In addition, if you have liver disease, you can, if you use the medicine properly for six months and if you are tested, it may be that the virus is gone", (participant 5, PLWHB, male, FGD 1).*

Within the HCW groups, although there was discussion around poor community HBV knowledge, they also noted a significant knowledge gap amongst HCWs prior to the STRIKE-HBV study. They felt STRIKE-HBV had given them confidence in HBV management:

**Table 1. Summary of the different themes arising from focus group discussions (FGDs) with people living with hepatitis B virus infection (PLWHB) and healthcare workers (HCWs) caring for PLWHB at Kilifi County Referral Hospital (KCRH) in Kilifi, Kenya. HBV – hepatitis B virus.**

| Knowledge and understanding | How lives have changed | Current care available | Barriers to care | How to reduce transmission and improve care |
|---|---|---|---|---|
| - Ongoing confusion about medication duration after the initial 6-month period.<br>- Poor community knowledge with no local language word for HBV<br>- HCWs poorly informed prior to STRIKE-HBV and incorrect information still given in other facilities<br>- PLWHB concerns about being 'bewitched' and visiting traditional healers to be 'cured'.<br>- Confusion with HIV. Many PLWHB having to 'justify' they don't have HIV. | - Unable to travel abroad due to HBV diagnosis.<br>- Reduced alcohol intake, better self-care.<br>- Reduced freedom in the evenings due to medication timing.<br>- Concerns about dying. | **Benefits**<br>- Adherence counselling and psychosocial support provided at KCRH.<br>- Staff at KCRH educate partners and family.<br>**Concerns**<br>- Testing not free of charge outside of STRIKE-HBV.<br>- No vaccination other than routine infant schedule.<br>- Smaller facilities have the correct provisions but feel unable to provide HBV care due to lack of knowledge | **Economic factors**<br>- Cost of transport<br>- Lack of decentralised clinics<br>- Living in poverty<br>- Loss of work<br>- Difficulties getting time off work to attend<br>**Stigma and lack of education**<br>- Self-stigma and community stigma<br>- Concerns from others about transmission<br>- Unhappy about HIV/HBV service integration due to increased stigma.<br>**Others**<br>- Lack of freely available testing and vaccination. | - Education about safe sexual practices.<br>- Reduce sharing of household implements (razors/toothbrushes)<br>- Provision of vaccination including birth dose vaccine.<br>- Community educational campaigns.<br>- TV and radio campaigns, advocacy by prominent society members and the ministry of health.<br>- Better availability of free testing, particularly in antenatal women. Testing available in communities.<br>- Decentralised clinics for those who want them, and education of more HCWs.<br>- Specific hepatitis clinics, separate from HIV. |

*"At first, I didn't know how to give the treatment. I used to just guess. I didn't know how long I should be giving this medication. I didn't even have the knowledge of package information to give the patient but right now I can stand comfortably with confidence and tell the patient what he or she is going to take and for how long the treatment should be and what he or she should do." (participant 7, HCW, male, FGD 2)*

**Perceptions and naming of HBV in the community.** When discussing a local language description of HBV, participants from both groups stated it was an unknown disease, or only described by the symptoms:

*"But this hepatitis B virus is unknown, let's say it has no name. People don't know, they only know AIDS", (participant 1, PLWHB, female, FGD 2).*

*"In Kilifi they just describe the complications, when you have jaundice they say "yellow", when you have ascites they say "mwamimba" [pregnancy], only that." (participant 1, HCW, female, FGD 2).*

Others reported HBV infection was called *'cancer'*. Several people mentioned that they had heard of HBV as it was causing illness in their village, and perceived it to be a death sentence:

*"I knew it [HBV] was killing people in the village. You don't know if he is starting [to get unwell] but you will only know that something has reached. That is, they don't get medicine, that is, there is no nursing, it is only when he is taken to the hospital, when he begins to be treated, then death has already arrived" (participant 6, PLWHB, male, FGD 1).*

**Beliefs in bewitchment and alternative treatments.** Several respondents in both HCW and PLWHB groups mentioned that PLWHB were sometimes thought to be bewitched, and they would go to traditional healers to be cleansed and take herbal medications instead of tenofovir, or to church to pray to be cured:

*"…those [with HBV] who have complications, or have ascites, in local language or native language we call it "mwa-mimba" [pregnant] and so they say that they have been bewitched and somebody has thrown something in their stomach and now like you are like a man but you are "pregnant". (participant 5, HCW, female, FGD 2).*

### ii) Confusion with HIV and stigma spillover

Both PLWHB and HCWs discussed the confusion between HBV and HIV, particularly given that care is received in the same clinic, and the medication is the same. Many PLWHB found it difficult to accept they did not have HIV, and felt they needed to 'prove' they didn't have HIV by going with family or partners to be tested:

*"…one day my wife came and explained to me, "I heard some people say that you have AIDS." I told her, "No, I don't have AIDS and if you want to know, let's go, let's all go and be tested" (participant 5, PLWHB, male, FGD 2).*

*"… [many] people who don't understand and then once they get the results and then they are being escorted to this building CCC, majority don't get to understand clearly the relationship like we are telling them it's not HIV and we are giving them ARV's (anti-retroviral therapy)." (participant 1, HCW, female, FGD 2).*

### iii) Post-diagnosis transformations: emotional distress, lifestyle changes and altered life plans

**Reactions to diagnosis.** Many PLWHB were shocked at their diagnosis and felt anxious as they had little knowledge of HBV. HCWs reported many newly diagnosed people thought they were dying, and associated HBV only with end stage disease. Counselling helped alleviate anxiety and was felt to be very important by both PLWHB and HCWs. Several PLWHB mentioned that once they got used to their diagnosis, they realised they could continue to take the medication and live just like anyone else. Reactions of friends and family varied. Some were very supportive, went to get tested themselves and encouraged participants to continue medication, but some partners left after the diagnosis.

**Lifestyle changes.** Almost all participants commented that their lives had changed since HBV diagnosis, sometimes in beneficial ways, some less so. One male participant had considerably cut his alcohol consumption due to concerns about his liver:

*"Even that time I came to be tested, I came already drunk…I have been told "The liver's job is this and this and this and when you eat, the food is processed like this and this and this in the stomach"... I saw that there is a very important knowledge…. I stopped drinking alcohol; I continue to take medicine. So I see there is a big change. That is, since I have been drunk all the time and this time, I can be sober, I'm grateful", (participant 1, PLWHB, male, FGD 1).*

Other participants perceived that they had less freedom, needing to be home from work at a particular time and not engaging in social activities due to concerns about taking their medication:

*"But since I use these drugs, I always tell my friends that I can't walk at night. Because when it reaches eight o'clock, I have to take medicine" (participant 7, PLWHB, female, FGD 1).*

**Changes to life plans.** Many women diagnosed with HBV had been tested as a condition for travelling abroad for work, and their diagnosis had not been properly explained to them which caused a lot of distress.

*"He [employment agent] refused to tell me because he said [the problem was] either liver or kidney. I told him, "Now there is no disease that is good and there is no disease which is bad, you can just explain it to me." He said, "Just go to the doctor and get tested or go and get an x-ray and he [the other doctor] will explain more, but I won't explain it to you."" (participant 3, PLWHB, female, FGD 2).*

**Changes in symptoms.** Many PLWHB had non-specific symptoms that improved with treatment such as tiredness, stomach pains and swelling, nausea, itching, heartburn and constipation:

> "I was given about five days [of medication]. I came back, I came and was given for two weeks, I came back. But since then, I see changes myself even in my body. I don't get hurt a lot and I don't get a lot of fatigue". (participant 7, PLWHB, female, FGD 1).

> "I was diagnosed with this disease, and I just saw that I was already dying. But when I got this treatment, I am fine now because even when I was sleeping and I could feel my stomach like water going this way, when I turned over, I could feel like water coming back this way and I said, "My stomach has started to swell too." But this time when I started this treatment, that issue stopped and the anxiety of saying that I was dying because I had not received [education]. This time I have received [education]. Now I know that if I use these drugs properly, I can live normally." (participant 5, PLWHB, male, FGD 1).

### iv) Current provision of HBV care in Kilifi

We discussed with HCWs what care was available for HBV both in KCRH and further afield, and how this had changed since the STRIKE-HBV study. Adherence counselling and medication had always been available at KCRH free of charge for HBV, however as described above, HCWs reported inadequate knowledge about medication and a lack of confidence when prescribing. In Kilifi County, diagnosis was most often made in private facilities prior to travel abroad for work, and patients were referred to KCRH for management as it was known to have an established hepatitis clinic. Occasionally people would be able to pick up their medication from smaller health centres, but only after education and service establishment by HCWs from KCRH:

> "So, there were no services [no hepatitis clinic] there, but we had to recreate the services for her to get the drugs. So, some have opted to go to a smaller health centre, and some to two sub-county hospitals, and so we have some sites which are offering [HBV treatment] but the diagnosis was done here." (Participant 7, HCWs, male, Group 2).

HCWs still perceived that peripheral facilities required more support to provide HBV care. Many health centres and dispensaries had HBV medication available along with adherence counsellors as they cared for those living with HIV, yet patients were still referred to KCRH:

> "…as much as they wish to go there, some facilities have limited knowledge on client follow up and now it becomes a challenge, but we have clients who would prefer to take medication from a nearby facility. I have had two who have been returned [to KCRH from peripheral facilities] for further management because the dispensary is not well equipped to manage hepatitis clients." (FGD 2, HCWs, female, participant 1).

Since STRIKE-HBV, HCWs reported they were able to provide better informed psychosocial support for PLWHB. They felt empowered to encourage PLWHB to talk to partners and family about testing and to be ambassadors and educate the community about HBV. Some HCWs were sharing their improved HBV knowledge with colleagues in peripheral facilities.

### v) Barriers to HBV care and how these can be reduced

**Economic Factors.** Economic factors seemed to be the biggest barrier to receiving care. Firstly, the cost of transport to attend a distant hospital-based clinic was mentioned by both HCWs and PLWHB. To counter this cost, many people felt HBV medication should be available at the local dispensary:

*"Because the rest of us are far away [from the hospital], but if we find our local dispensaries nearby, these local medicines, the way we are known, they are placed there. It will be easy even for us because we won't have to pay fare"*, (participant 2, PLWHB, female, FGD 1).

*"…they come from far and first they have the transport issues and secondly they are living in abject poverty and so most of them they cannot make it to Kilifi and so at least if we have the resources [medication] and we dispense the resources [medication] nearer to them then it will be better."* (participant 7, HCWs, male, FGD 2).

Several PLWHB mentioned the need to plan to make sure they had fare to attend the clinic. Along with fare, the timing of clinic appointments had little flexibility, and often the patients experienced long waiting times. This was difficult with employment, with men in particular mentioning they struggled to get days off work leading to lost earnings:

*"So the challenge always arises because maybe your boss refuses to agree with you, maybe it [the clinic return date] can be balanced until you get the chance to come take medicine or a meeting like this. You may find that your boss telling you, "Now you might not get paid today, or we will remove this day from your rest days, we will cut it there".* (Participant 5, PLWHB, male, FGD 2).

The wish for decentralised clinics however was not universal. Several people in both the patient and HCW groups mentioned that (despite the potential higher cost) some PLWHB would prefer to go to a distant clinic (where they were not known) to reduce stigma. However, sometimes asking for money for transport caused problems with questions about where they were going:

*"…many of them still have stigma so they are thinking like if I am seen there, I will be isolated, so many prefer to go to those far away hospitals that no one knows, very few said they were referred to go to a nearby hospital."* (Participant 4, HCW, female, FGD 1).

*"[If I ask a friend]: Help me with 200 shillings ($1.5) so that I can go somewhere and then I will give you back." He asks, "Where do you want to go?" What do you want to do with it?" Now that's where I see the difficulty."* (Participant 5, PLWHB, male, FGD 1).

**Community and self-stigma.** Community and self-stigma was raised as an issue by all focus groups, both men and women. PLWHB felt they were being talked about, and others distanced themselves for fear of infection:

*"I can't eat with them at the same table, it is as if I will infect them, I should have my own bowl and sit aside"*, (participant 3, PLWHB, female, FGD 1).

As noted earlier, HIV and HBV services are currently integrated in KCRH and all clients are seen together in the same clinic. PLWHB revealed tensions around this, with some feeling they should have a separate clinic from those living with HIV, partly due to stigma when seen entering the HIV clinic, but also due to long wait times when being seen together. Some people felt those living with HIV were prioritised, although there was also agreement that separating those with HBV and HIV could increase stigma, and they should all be together because they had more in common than differences:

*"I see they should just be together because we cannot separate them [HIV patients]. We all say we are birds of the same colour, we are going in the same direction. It means that if we separate them, it is like we are bringing that stigma and we don't want it to be like that"*, (participant 2, PLWHB, male, FGD 2).

## vi)  Reducing transmission

When asking PLWHB how to reduce HBV transmission, practical ways were discussed to reduce contact with the virus including safe sexual practices, wearing gloves and reducing the sharing of instruments that spread HBV such as needles, razors and toothbrushes. A few participants mentioned vaccination, but did not identify groups they felt could benefit from vaccination, or how vaccination should be offered.

Access to free testing was mentioned by several PLWHB, ideally as part of routine screening on hospital attendence, like HIV. There was also discussion of the benefits of antenatal testing for HBV, and how this should be routine:

> *"… There should be free testing, random or all over, everyone to be reached. I can say here in Kilifi, the people of Kilifi are lucky because even I am lucky because of your research and the services that are provided here." (participant 3, PLWHB, female, FDG 1).*

> *"…perhaps if the pregnant mother it is her first time, she must come with her husband so that they can be tested for AIDS, and the likes. Now I was thinking it would also be good if this hepatitis B was placed in that category, when they come to be tested, if she has come to the clinic, being tested for the HIV virus, they should also test the Hepatitis B virus…", (participant 6, PLWHB, male, FGD 2)*

## vii)  Improving education

A common theme that ran through all the discussion questions was the need for education and community awareness. This was seen as crucial to reduce stigma, transmission, and improve testing and diagnoses. Suggestions for education initiatives included television or radio, and respected community members such as pastors and chiefs:

> *"So you find when people already know that [HBV can be dangerous], if the pastors also intervene, after explaining, now you will get a large group that say, "Uh, let me look at myself." And there I think we will find many people will volunteer [for HBV testing]. You can't go to someone's house and grab his shirt and bring him here, no, it's up to him to decide and say, "At my own free will, let me check."", (participant 6, PLWHB, male, FGD 2).*

Several HCWs felt the community health promoters (CHPs) could play a pivotal role in HBV sensitisation and treatment adherence. They mentioned how CHPs are funded to visit people living with HIV at home but not those living with HBV:

> *"I feel the CHP should also be brought on board on issues of Hepatitis B sensitisation and treatment adherence. Because you will get CHP bringing a referral or a patient who had defaulted treatment for [HIV treatment] but they will never do follow up for a client with Hepatitis." (Participant 1, HCW, female, FGD 2).*

Several participants suggested continuing education for PLWHB through expert-led peer support sessions, along with including the immediate family and community. Others suggested if those living with HBV felt able to reveal and discuss their diagnosis, they could also be the ones to spread awareness:

> *"I think it would be better for those of us who have the disease to not be ashamed. I explain that I was diagnosed with that disease [HBV] and that I use those drugs, I tell the people there to go to the hospital, not to wait until the organisation reaches the grassroots, it will be too late", (participant 4, PLWHB, male, FGD 2).*

## Discussion

We set out to explore the experiences of both PLWHB and HCWs providing HBV care in Kilifi Kenya aiming to understand HBV knowledge, impact of HBV on people's lives and barriers to accessing care. Our findings, summarised in Table 1, add to the limited literature exploring the lives of PLWHB in WHO-AFRO countries. Several studies have evaluated different mechanisms to try and improve HBV care, however very few have included the patient voice, or discussed with HCWs involved with patient care when planning interventions, making this a valuable addition to HBV literature [20,21]

Financial barriers have also been highlighted elsewhere [7,22,23], and can lead to catastrophic costs for individuals and families living with HBV [24]. In our study, travel costs to the hospital clinic were seen as most prohibitive, whereas in studies from Burkina Faso and Ghana, costs of tests, monitoring and medications were mentioned more commonly [7,23]. This may be because in our study testing and nucleoside analogue therapy were provided free at the point of care in KCRH which is not the case in many other countries.

Many studies both from within WHO-AFRO and elsewhere have raised concerns around lost employment [25–27] as in our study, and this was particularly common amongst men. Lack of childcare was mentioned as a significant barrier to care in some studies outside of WHO-AFRO [26]. This was not mentioned at all in our study or others from Ghana or Burkina Faso [7,23] possibly due to the different family structures in countries in WHO-AFRO compared to those in the Global North, with less reliance on parents being the sole care givers.

The concerns raised by our participants around the integration of HBV and HIV care particularly related to increased stigma has been discussed previously [28]. The utilisation of existing HIV services is one of the key components hypothesised to enable decentralisation of HBV care. However, we suggest that caution is needed and whilst decentralised clinics should be available to those who want services closer to home, extensive community education and sensitisation should be done before this is implemented and access to services further afield should be maintained. Our finding of confusion between HIV and HBV was also described in Burkina Faso where HBV was closely associated with HIV when explained to patients [23]. Although this was seen as a negative association, the good understanding of HIV transmission routes did help explain HBV transmission. In Burkina Faso, those diagnosed with HBV outside of a hospital setting were often given poor information around HBV, similar to several of our PLWHB who were diagnosed pre-travel.

An interesting finding in our study was that many PLWHB felt they were restricted in the evenings due to taking medication, having to be home from work at a particular time or being unable to meet friends. This has been described in HIV literature as people are more likely to take their medication at home and do not want to carry tablets with them due to fear of discovery [29]. It may also be that the HBV clinic has given instructions around when to take medication related to food to reduce side effects, and this is most convenient in the evenings. Education from the HBV clinical team around the flexibility of medication timing could be helpful, so PLWHB understand it can take be taken at any time of day as long as this is consistent and does not need to interfere with any other social, domestic or work commitments.

Several participant suggestions align with national Kenyan HBV management guidelines such as antenatal HBV testing and provision of vaccination to close family contacts [30], however these are not universally incorporated into care. To continue strengthening HBV care at KCRH and across the County, education should be a funding priority as it was seen as one of the most important initiatives to increase diagnoses, drive down stigma and improve HBV care pathways. As HCWs frequently transfer between facilities, even localised educational initiatives have the potential to disseminate knowledge more broadly.

## Caveats and limitations

We only included PLWHB in this study who were already linked to care and receiving treatment, and both HCWs and PLWHB had been involved in previous research and have engaged with a peer support group. We also asked people to volunteer to participate in these FGDs. These groups are likely to have better knowledge and be more engaged in care than those not involved in such a study. Those volunteering may have had specific experiences when living with HBV

or treating those with HBV making them more likely to participate. It is also possible that this previous experience influenced thoughts expressed in these FGDs. We included people from a limited geographical area and only a small number of HCWs and PLHWB. We did not continue data collection until saturation due to time and financial constraints. To fully understand care barriers, we need to firstly undertake more FGDs within this group to ensure themes are saturated, and to ensure all viewpoints are represented. We then need to undertake similar discussions with PLWHB who have been lost to follow up, those not offered treatment and include community groups not tested or living with HBV; ideally a future study would expand to include PLWHB and HCWs accessing healthcare services both in Kilifi and further afield.

## Conclusions

Community education and decentralisation of care for PLWHB should be prioritised to reduce stigma, enable easier access to care, reduce travel times and costs to reduce health inequities and support wider roll-out of treatment. Community co-design is critical to ensure interventions are appropriate, accessible, acceptable and affordable, and can be sustained over time.

## Supporting information

**S1 File.  Topic guides for focus group discussions with people living with hepatitis B virus infection (HBV).** 18th June 2024.
(DOCX)

**S2 File.  Topic guides for focus group discussions with healthcare Workers caring for those living with HBV.** 14th November 2024.
(DOCX)

**S1 Table.  Changes in HBV care in Kilifi County due to the STRIKE-HBV study.**
(DOCX)

**S3 File.  Inclusivity in global research.**
(DOCX)

## Acknowledgments

We would like to thank all the PLWHB and HCWs who took part in these FGDs, along with the CCC team for facilitating the use of their clinic space to undertake this work. Without their enthusiasm and dedication to improving care for those living with HBV, this would not have been possible.

## Author contributions

**Conceptualization:** Louise O Downs, Mwanakombo Zaharani, Oscar Chirro, Benson Safari, Janet Seeley, Philippa C Matthews, Nadia Aliyan.

**Data curation:** Louise O Downs, Juliet Odhiambo, Mwanakombo Zaharani, Oscar Chirro, Benson Safari.

**Formal analysis:** Louise O Downs, Juliet Odhiambo, Philippa C Matthews, Nancy Kagwanja.

**Funding acquisition:** Louise O Downs, Nadia Aliyan.

**Methodology:** Louise O Downs, Juliet Odhiambo, Mwanakombo Zaharani, Oscar Chirro, Benson Safari, Janet Seeley, Philippa C Matthews, Nadia Aliyan, Nancy Kagwanja.

**Project administration:** Louise O Downs.

**Resources:** Nadia Aliyan.

**Supervision:** Juliet Odhiambo, Janet Seeley, Philippa C Matthews, Nadia Aliyan, Nancy Kagwanja.

**Validation:** Nancy Kagwanja.

**Writing – original draft:** Louise O Downs.

**Writing – review & editing:** Juliet Odhiambo, Mwanakombo Zaharani, Oscar Chirro, Benson Safari, Janet Seeley, Philippa C Matthews, Nadia Aliyan, Nancy Kagwanja.

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
