## [Decision Letter · Decision Letter 0]

15 Jun 2025

PGPH-D-25-00554

“I think it would be better for those of us who have the disease not to be ashamed”. Insights from people living with chronic hepatitis B virus infection and healthcare workers providing HBV care in Kilifi, Kenya.

Dear Dr. Downs,

Thank you for submitting your manuscript to PLOS Global Public Health. After careful consideration, we feel that it has merit but does not fully meet PLOS Global Public Health’s publication criteria as it currently stands. Therefore, we invite you to submit a revised version of the manuscript that addresses the points raised during the review process.

The manuscript has been evaluated by two reviewers, and their comments are available below.

The reviewers have raised a number of major concerns. Both reviewers mentioned limitations to address and aspects of the methods they could be described more clearly. There were also recommendations to the flow of particular sentences in the manuscript.

Could you please carefully revise the manuscript to address all comments raised?

We look forward to receiving your revised manuscript.

Kind regards,

Katherine Demi Kokkinias, Ph.D.

Staff Editor

Journal Requirements:

2. Please send a completed 'Competing Interests' statement, including any COIs declared by your co-authors. If you have no competing interests to declare, please state "The authors have declared that no competing interests exist". Otherwise please declare all competing interests beginning with the statement "I have read the journal's policy and the authors of this manuscript have the following competing interests:"

3. In the online submission form, you indicated that Full recordings and transcripts in both English and Kiswahili of these FGDs are held on a secure server at KWTRP but could be accessed by other studies on request to dgc@kemri-wellcome.org and approval by the data governance committee. For the purpose of Open Access, the author has applied a CC-BY public copyright license to any author accepted manuscript version arising from this submission. This manuscript was written with the permission of Director KEMRI CGMRC.

3. Uploaded as supplementary information.

Reviewers' comments:

Reviewer's Responses to Questions

**Comments to the Author**

1. Does this manuscript meet PLOS Global Public Health’s publication criteria?

Reviewer #1: Yes

Reviewer #2: Yes

2. Has the statistical analysis been performed appropriately and rigorously?

Reviewer #1: N/A

Reviewer #2: N/A

3. Have the authors made all data underlying the findings in their manuscript fully available (please refer to the Data Availability Statement at the start of the manuscript PDF file)?

Reviewer #1: Yes

Reviewer #2: Yes

4. Is the manuscript presented in an intelligible fashion and written in standard English?

Reviewer #1: Yes

Reviewer #2: Yes

Reviewer #1: This is a well-written study that explores the understanding of HBV infection at both individual and community levels, it examines how the lives of PLWHB have changed since their diagnosis and investigates barriers to care while considering strategies to mitigate them. The study’s objectives are clearly defined, and it effectively delivers on its aims.

However, I have two queries:

1. How did you ensure that responses from focus group discussions were not influenced by the interviewer’s expectations? As noted in the study’s limitations, both healthcare workers and PLWHB have participated in previous research and engaged in peer support groups. How do you account for the potential influence of prior discussions and engagements on their responses? In this regard, would it be beneficial to include the interview questions as supplementary material and reference them in the methods section. This could help demonstrate how the questions encouraged participants to elaborate on their own experiences.

2. Regarding participant recruitment, the study mentions that participants were purposively sampled to ensure diversity in age and experience. However, it is unclear whether data collection continued until saturation was reached and whether the selected groups effectively contributed to achieving this. Could this be clarified or acknowledged as a limitation?

Reviewer #2: Review comments

General observation: This manuscript offers a valuable contribution to the literature on hepatitis B virus (HBV) care in the WHO-AFRO region, particularly by highlighting the perspectives of both people living with HBV (PLWHB) and healthcare workers (HCWs). The study effectively combines qualitative insights to provide understanding of the challenges faced by PLWHB in accessing care, as well as the barriers encountered by HCWs in delivering adequate treatment. The authors offer actionable recommendations for improving HBV care, particularly through community education, decentralization of services, and addressing stigma. The inclusion of both PLWHB and HCWs’ voices enriches the study and makes it a significant addition to the ongoing discussions around HBV management in low-resource settings. Overall, this is a well-executed study that provides practical suggestions for improving care pathways and offers important insights for policymakers and health systems in the region.

The following comments can help improve the manuscript.

Introduction:

• Can the flow between the Kenya-specific situation and broader African context be made smoother for better readability? The paragraph transitions are abrupt. The shift from Kenya’s situation (lines 61–67) to WHO-AFRO and stigma (lines 69–77) feels disjointed.

• Briefly elaborate on the HBV stigma survey’s findings or relevance, instead of mentioning it vaguely.

• The author may consider moving the methodological details (platform building, SRQR compliance) to the Methods section for cleaner structure. The methods are leaking into the introduction.

Methods

Context

• Could you provide more information on why there is no routine laboratory or imaging assessment for HBV treatment eligibility at KCRH? This is important for understanding the limitations in the clinical setting, otherwise readers may wonder why such assessments are not available or why this is the standard practice at the hospital.

• The description of HBV care at KCRH could benefit from more detail. It would be useful to explicitly state whether the Comprehensive Care Clinic (CCC) only handles HIV and HBV cases or if it serves other infectious diseases as well. This would provide a clearer understanding of how HBV care fits within the broader healthcare framework at KCRH.

Data collection

• The dates (e.g., "18th June 2024") should be consistently formatted. Consider either writing the month fully ("18 June 2024") or using the same format throughout.

• Could the selection of participants during peer support sessions lead to any potential bias in the representativeness of PLWHB, and how might this affect the findings?

Data analysis

• You mention that LD developed the coding framework and refined it through discussion with the team. It would strengthen your credibility to briefly explain how disagreements or differences in interpretation were handled.

• Was coding refined iteratively as new data came in (i.e., were earlier codes updated after later FGDs)? If yes, briefly stating this would highlight reflexivity and responsiveness to the data.

Results

Knowledge and understanding of HBV

• Paragraphs from lines 168 to 214 are dense. Breaking them into clear sub-sections or signalling sub-themes (e.g., understanding of HBV transmission, perceptions and naming of HBV in the community, Beliefs in bewitchment and alternative treatments) would help readability.

• Some redundancy in quotes: A couple of quotes echo the same point (e.g., HBV perceived as fatal/unknown). One or two could be trimmed to avoid repetition and improve narrative flow.

Comparison with HIV

• The second theme "Comparison with HIV" is accurate but broad. The author may consider a more descriptive title like: "Confusion with HIV, stigma spillover, and funding disparities between HBV and HIV" or anything deemed better.

How lives have changed since HBV diagnosis

• "How lives have changed since HBV diagnosis" is accurate but can be more analytic. Something like: “post-diagnosis transformations: emotional distress, lifestyle change, and altered life plans."

• The paragraph/section is currently long and hard to navigate. Breaking it into subthemes would greatly enhance clarity.

Barriers

Economic factors

• The last two quotes (on stigma and difficulty asking for fare) feel somewhat tacked on. A transition sentence summarizing the tension between proximity and stigma would smooth the flow.

Stigma and lack of education

• The first quote feels abrupt due to missing transition or setup. A lead-in sentence clarifying the context of self-stigma at home would improve flow.

Reducing transmission

• The idea that education is key is repeated several times. A concise synthesis paragraph summarizing these points once, rather than reiterating throughout, might improve readability.

Discussion

• Tables are labelled at the top

• Ensure consistency in how references are cited—e.g., use either "(23,24)" or "[23,24]" uniformly, based on your chosen style guide.

• Some sentences are long and slightly convoluted. Example:

"Several suggestions from participants here on how to improve care are already included in national Kenyan guidelines…"

Could be revised to:

"Several participant suggestions align with national Kenyan HBV management guidelines, such as antenatal HBV testing and vaccinating close family members…"

• Minor grammatical error on line 486

Limitations

• Fix typo in the abbreviation on line 504

Note: I have incorporated some comments in the PDF version of the manuscript.

Thank you and all the best.

**Do you want your identity to be public for this peer review?** For information about this choice, including consent withdrawal, please see our Privacy Policy

Reviewer #1: No

Reviewer #2: No

---

## [Decision Letter · Decision Letter 1]

4 Sep 2025

PGPH-D-25-00554R1

“It would be better for those of us who have the disease not to be ashamed”. Insights from people living with chronic hepatitis B virus infection and healthcare workers providing HBV care in Kilifi, Kenya.

Dear Dr. Downs,

Thank you for submitting your manuscript to PLOS Global Public Health. After careful consideration, we feel that it has merit but does not fully meet PLOS Global Public Health’s publication criteria as it currently stands. Therefore, we invite you to submit a revised version of the manuscript that addresses the points raised during the review process.

We look forward to receiving your revised manuscript.

Kind regards,

Helen Howard

Staff Editor

Journal Requirements:

Additional Editor Comments (if provided):

Reviewers' comments:

Reviewer's Responses to Questions

**Comments to the Author**

Reviewer #2: All comments have been addressed

Reviewer #3: (No Response)

publication criteria?

Reviewer #2: Yes

Reviewer #3: Partly

3. Has the statistical analysis been performed appropriately and rigorously?

Reviewer #2: N/A

Reviewer #3: N/A

4. Have the authors made all data underlying the findings in their manuscript fully available (please refer to the Data Availability Statement at the start of the manuscript PDF file)?

Reviewer #2: Yes

Reviewer #3: Yes

5. Is the manuscript presented in an intelligible fashion and written in standard English?

Reviewer #2: Yes

Reviewer #3: Yes

Reviewer #2: The authors have adequately addressed all my comments from the previous review, providing clear revisions and justifications where necessary.

Thank you.

Reviewer #3: The authors report the findings of a qualitative study detailing the experiences of PLWHB and HCWs who care for PLWHB from a single hospital in Kenya. Overall, the results are interesting and informative but this study is limited by its small sample size. Specific comments for improvement are available below.

Specific Comments

Overall:

- The writing style has many “run-on” sentences that could be shortened for brevity and clarity

- The current manuscript has well over 6,000 words. This needs to be reduced substantially to less than 4,000 words.

Introduction

- Sentence starting with “HBV has, however…” reads awkwardly. Consider something like “Despite this global burden, HBV has been neglected…”

- Sentence starting with “an estimated 4.2%...” is unclear. Do you mean that 4.2% of all people have HBV or that only 4.2% of all estimated patients with HBV have been diagnosed?

- Line 64: you state “little vaccination outside of the childhood immunisation series”. Most children who have received their childhood immunization series do not need additional doses. Do you mean that HBV vaccines are unavailable to adults who did not receive their childhood HBV series?

- Sentence line 70-71: please provide citation

- Sentence line 75-77: you already say something similar in lines 66-68

- Condense sentences from 77-82 into a single sentence

- Sentence 84-86: This sentence needs to be revised. I would start with “We aimed to describe the experiences of PLWHB and HCWs caring for PLWHB in Kilifi County Referral Hospital (KCRH) in Kenya, to help inform care delivery. We also sought to establish a scalable platform for qualitative data collection that would be useful in other settings.”

Materials and Methods

- I’m not sure you need a whole table 1 to describe the facilities in Kilifi County. This information does not seem directly relevant to your paper.

- Table 2 does not directly relate to what was conducted in this study. This could be presented in a supplement, but should not appear in the main body of the manuscript.

Results

- Table 3 should be explicitly referred to in the results section

- Consider cutting back significantly on the number of words in this section

Discussion

- Line 620-622: Earlier in the manuscript you state that not all participants preferred decentralization, because it could actually worsen stigma. So I think this sentence has to be edited to reflect that it should be decentralized for those who wish to have services closer to home.

**Do you want your identity to be public for this peer review?** For information about this choice, including consent withdrawal, please see our Privacy Policy

Reviewer #2: No

Reviewer #3: No

---

## [Editor Report · Decision Letter 2]

18 Sep 2025

“It would be better for those of us who have the disease not to be ashamed”. Insights from people living with chronic hepatitis B virus infection and healthcare workers providing HBV care in Kilifi, Kenya.

PGPH-D-25-00554R2

Dear Dr Downs,

We are pleased to inform you that your manuscript '“It would be better for those of us who have the disease not to be ashamed”. Insights from people living with chronic hepatitis B virus infection and healthcare workers providing HBV care in Kilifi, Kenya.' has been provisionally accepted for publication in PLOS Global Public Health.

Best regards,

Julia Robinson

Executive Editor